# SASS: SELF-ALIGNMENT WITH SEMI-SUPERVISED INSTRUCTION DATA GENERATION

## ABSTRACT

Instruction tuning is important in enabling Large Language Models (LLMs) to align with user expectations to complete various open-domain tasks. The success of instruction tuning depends on the availability of high-quality instruction data. Owing to the exorbitant cost of high-quality human annotation, recent works have been deeply engaged in the exploration of the utilization of LLMs to generate instruction data automatically. However, these methods carry potential risks arising from the usage requirements of powerful closed-source models, which strictly forbid the utilization of their outputs to develop machine learning models. In this paper, we focus on using open-source models to generate high-quality data for themselves instead of distilling from other powerful models and explore various semi-supervised instruction data generation strategies to empower the limited capabilities of open-source models. To further improve the diversity of generated instructions and the alignment between the generated instructions and selected responses, we also propose two techniques, i.e., instruction filtering and extract-then-generate strategy. Experiments on both the LongForm benchmark and GPT-4 evaluation confirm the effectiveness of our proposed strategies. Besides, we surprisingly find that even for the relatively weak models, the performance of generating data for themselves is on par with distilling from the more powerful model. We hope that more progress can be achieved in generating high-quality instruction data without using closed-source models.

## 1 INTRODUCTION

Instruction tuning has received a wide range of attention from the Natural Language Processing (NLP) community (Mishra et al., 2022; Wei et al., 2021; Sanh et al., 2021; Chung et al., 2022; Ouyang et al., 2022; Wang et al., 2022a). Through the utilization of various types of instruction data, LLMs can fully exploit the knowledge obtained in the pre-training stage, leading to the superior performance of generalization capability across unseen tasks. This ability to transcend task-specific limitations illustrates the potential of LLMs for Artificial General Intelligence (AGI).

The quality of data is paramount to the success of instruction tuning (Zhou et al., 2023; Chen et al., 2023). Existing instruction datasets are either created through manual annotation or generated by models. The former method involves the manual conversion of existing NLP datasets into an instruction format, but this approach is constrained by high cost and variable quality (Wang et al., 2022a; Honovich et al., 2022). The latter method, on the other hand, typically uses powerful closed-source models to generate instruction datasets (Wang et al., 2022a; Honovich et al., 2022; Taori et al., 2023; Köksal et al., 2023; Yin et al., 2023). Unfortunately, the usage requirements of powerful closed-source models usually restrict using their

outputs for developing machine learning models, e.g., OpenAI [1], Google [2] and Anthropic [3], introducing potential risks to existing model generation methods. Powerful open-source models have similar requirements (LLaMa2 [4]). Therefore, the development of a novel method to automatically generate high-quality instruction data without relying on closed-source models becomes imperative. However, fulfilling this demand is challenging due to the need for high-quality instruction data that meets multiple criteria including instruction diversity, response validity, and tight alignment between instruction and response. Due to the limited capabilities of open-source models, it is more challenging to address these problems.

To address the above problems, in this work, we systematically explore utilizing open-source models to generate instruction data for themselves instead of distilling from other LLMs. Limited by the relatively weak capabilities, in our preliminary experiments, open-source models cannot generate paired instruction data like powerful closed-source models. Therefore, we explore two main semi-supervised instruction strategies: generating responses for existing unsupervised instructions (self-training strategy) and generating instructions for unsupervised responses (instruction-generation strategy). Although the usage requirements claim that the supervised instruction datasets collected from closed-source models cannot be used to train other models, the input instructions of these datasets are written by users and not restricted by this clause. Therefore, we remove the responses generated from closed-source models and use the remaining instructions for the self-training. For the instruction-generation strategy, we collect unsupervised responses corpus from various document sources, e.g., webpages, books, and code.

The contributions of this work and most surprising findings are as follows:

- We systematically explore various semi-supervised instruction data generation strategies, which use open-source models to generate instruction data for themselves;

- We find that with the ensure of response validity, the instruction-generation strategy outperforms self-training significantly and is robust to the seed instruction data;

- We propose an instruction filtering strategy to ensure the alignment between the generated instructions and selected responses and an extract-then-generate strategy to improve the diversity of instructions. Experimental results confirm the effectiveness of our proposed strategies;

- With the help of our proposed strategies, for the relatively weak models, the performance of generating data for themselves is on par with distilling from the more powerful model.

## 2 RELATED WORK

In recent years, instruction tuning has gained significant attention, which can enable LLMs to generalize to unseen tasks. In this section, we divide existing instruction datasets into two groups: manual annotation instruction datasets and model generation instruction datasets.

**Manual Annotation Instruction Datasets** For the purpose of generalizing LLMs to unseen tasks, instruct tuning needs a large number of instruction data with various task types. The most straightforward way to achieve this goal is to collect from existing NLP datasets. Therefore, early instruction datasets are collected from large-scale existing NLP task datasets and transformed into instruction formats with manual written templates, e.g., Natural Instructions (Mishra et al., 2022), Flan 2021 (Wei et al., 2021), and T0 (Sanh et al., 2021). With the scaling law of instruction data that more task types and data can improve the performance

---

[1] https://openai.com/policies/terms-of-use

[2] https://policies.google.com/terms/generative-ai

[3] https://vault.pactsafe.io/s/9f502c93-cb5c-4571-b205-1e479da61794/legal.html#terms

[4] https://github.com/facebookresearch/llama/blob/1a240688810f8036049e8da36b073f63d2ac552c/LICENSE

of instruction tuning (Xu et al., 2022), subsequent works collect more data from existing datasets to expand tasks to the thousands, e.g., ZeroPrompt (Xu et al., 2022), Super-Natural Instructions (Wang et al., 2022b) and OPT-IML (Iyer et al., 2022). To further enrich data sources, xP3 (Muennighoff et al., 2022) adds multilingual instruction tuning; Flan Collection (Chung et al., 2022; Longpre et al., 2023) collects chain-of-thought data and samples some data to be organized in the format of in-context learning. Gu et al. (2022) leverages human-written rules and templates to convert unsupervised data into the instruction data format to further raise the number of data resources. In order to align with human requirements in realistic scenarios rather than collecting instruction data from existing NLP datasets, some recent works focus on collecting instruction data from realistic scenarios (Ouyang et al., 2022; Databricks, 2023).

**Model Generation Instruction Datasets**   Limited by the high cost of manual annotation, it is unaffordable to collect high-quality instruction data manually. Therefore, model generation models focus on generating instruction data automatically. Existing methods have explored the effectiveness of employing large language models to generate high-quality instruction data. Specifically, by providing multiple seed instruction-tuning tasks as prompts, Self-Instruct (Wang et al., 2022a), Unnatural Instruction (Honovich et al., 2022), and Alpaca (Taori et al., 2023) can follow the format of the given seed tasks to generate instruction data automatically. Subsequently, there is a line of work to improve the quality of generated data further. Specifically, Xu et al. (2023) focuses on improving the task complexity of instruction data; Jiang et al. (2023) proposes an adversarial distillation framework to utilize the feedback of student models; Ding et al. (2023) utilizes two separate LLM APIs to generate conversational instruction data. Besides, relying solely on LLMs to generate both instruction and outputs may generate low-quality data (Köksal et al., 2023; Yang et al., 2023b). To improve the quality of generated instruction data, LongForm (Köksal et al., 2023) is the pioneering work leveraging LLMs to generate the instruction based on an existing corpus. Dynasaur (Yin et al., 2023) uses LLMs to generate instruction data with the help of the meta-information of existing NLP datasets. RefGPT (Yang et al., 2023b) handles the hallucination problem by constraining the LLMs to rely on the provided reference instead of generating dialogues based solely on their own knowledge.

## 3   PRELIMINARY STUDY

In this section, we compare the performance of two main categories of semi-supervised instruction data generation strategies: using instructions to generate responses, i.e., Self-Training (ST), and using responses to generate instructions, i.e., Instruction-Generation (IG).

### 3.1   SEMI-SUPERVISED INSTRUCTION GENERATION STRATEGIES

**Self-Training**   Due to the usage requirements of powerful closed-source models, the output of closed-source models cannot be used to fine-tune other models. However, in the instruction datasets collected from the closed-source model, the input instructions are written by the user, and utilizing these instructions will not violate usage restrictions. Therefore, we explore the self-training strategy to utilize these datasets. Specifically, we first use seed human-annotated instruction data to fine-tune one open-source model and obtain a seed instruction-following model. Then, the seed instruction-following model is used to generate responses for the collected human-written instructions. Finally, we pair the generated responses with corresponding instructions to fine-tune the final instruction-following model.

**Instruction-Generation**   In the instruction-generation strategy, we generate instructions for the selected documents. Specifically, we first reverse the input and output of human-annotated instruction data to train a seed instruction generation model. Then, we use the seed instruction generation model to generate appropriate instructions for the selected unsupervised documents. Finally, we use these generated instruction data to obtain the final model.

### 3.2 STUDY SETUP

**Data Source** For the seed human-annotated instruction data, we use two data sources: Lima (Zhou et al., 2023) and Dolly (Databricks, 2023). In the self-training experiments, we use the input instructions of ShareGPT (Chiang et al., 2023) as the unsupervised instructions. Specifically, we only select the user input in the first conversation turn as unsupervised instructions. In the instruction-generation experiments, we use documents in the minipile (Kaddour, 2023) as the candidate responses.

**Implementation Details** We implement all models with the open-source toolkit *Transformers*[5] Wolf et al. (2020). In the fine-tuning stage, we follow Taori et al. (2023) use AdamW (Loshchilov & Hutter, 2019) optimizer and set the learning rate to 1e-5, batch size to 32, learning rate warmup ratio to 0.03, and perform training for 3 epochs. We utilize LLaMA-7B (Touvron et al., 2023) as the backbone model and update all parameters during the fine-tuning stage. All the fine-tuning experiments are performed with 80GB NVIDIA A100 GPUs. During the instruction generation stage, we employ Nucleus Sampling (Holtzman et al., 2019) to generate diverse data. Specifically, we set the Top-p to 0.9, the Top-k to 40, and the temperature to 0.7. For all data augmentation experiments, we augment 50,000 samples. We follow Taori et al. (2023) to design the training template:

> ***Below is an instruction that describes a task, paired with an input that provides further context.***
> ***Write a response that appropriately completes the request.***
> ***### Instruction:{instruction}***
> ***### Input:{input}***
> ***### Response:***

For the instruction-generation model, we set the instruction as "Write an appropriate instruction for the given text." and fill in the unsupervised document as "input".

### 3.3 AUTOMATIC EVALUATION BENCHMARK

We use the comprehensive benchmark LongForm (Köksal et al., 2023) as the automatic evaluation benchmark. Long-Form has a total of 2,045 examples, including various task types, e.g., question answering, grammatical error correction, story generation, text summarization, and email writing. Specifically, LongForm consists of 8 sub-tasks: C4, Wikipedia, Stack Exchange, WikiHow, Supernaturalinstructions, BEA-GEC, Enron, and BigBench. For the automatic metrics, we use the original evaluation metrics (Köksal et al., 2023), i.e., METEOR (Banerjee & Lavie, 2005) score for LongForm. For generation, we follow Köksal et al. (2023) to perform nucleus sampling with p = 0.9 for all models.

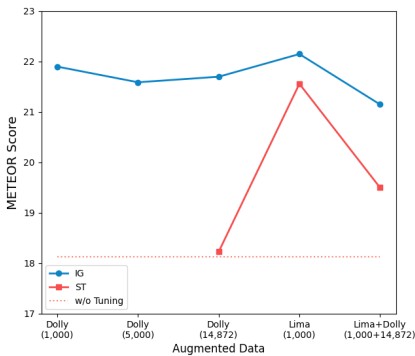

Figure 1: The average performance of the LLaMa-7B model with Self-Training (ST) and Instruction-Generation (IG) strategies on the LongForm benchmark. We study the effects of different seed instruction data.

### 3.4 RESULTS AND ANALYSIS

Figure 1 shows the results of self-training and instruction-generation. We can find that the instruction-generation strategy can outperform the self-training strategy significantly. We hypothesize that generating appropriate responses for one given instruction is more difficult than vice versa,

---

[5]https://github.com/huggingface/transformers

and the instruction-generation strategy can ensure the validity of responses. Besides, we find that whether using different data sources or randomly sampling different amounts of data under the same data source, the results of the instruction-generation strategy just show little fluctuation, which shows seed instruction data has little impact on the instruction-generation strategy. This phenomenon further confirms our hypotheses that it is easier to generate appropriate instructions for the given responses.

## 4 METHOD

In this section, due to the superiority of the instruction-generation strategy over the self-training strategy, we propose two strategies to improve the performance of the instruction-generation strategy further. The illustration of the overall instruction data generation framework is shown in Figure 2.

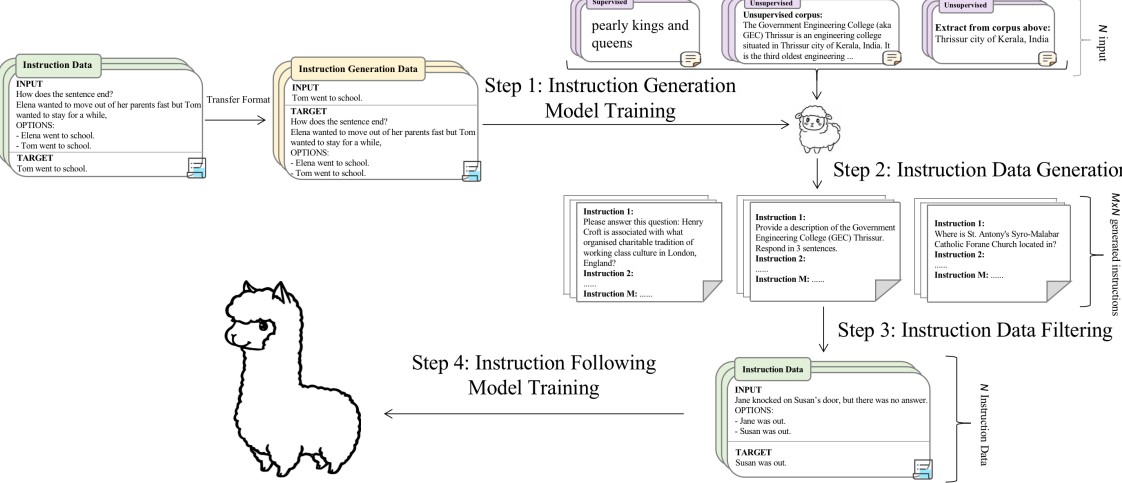

Figure 2: An illustration of our instruction data generation framework. Specifically, in **Step 1**, we reverse the instruction and output of original instruction data to train an instruction generation model; in **Step 2**, we use the fragments collected from the existing corpus as the output and use the instruction generation model to generate candidate instructions for these selected outputs; in **Step 3**, we propose a novel instruction filtering strategy to select the most appropriate instruction; in **Step 4**, we use the filtered instruction data to train the final instruction-following model.

### 4.1 INSTRUCTION FILTERING STRATEGY

In order to improve the alignment between generated instructions and selected responses, we propose an instruction filtering strategy. The motivation of this strategy is that if the generated instruction is the most appropriate instruction for the selected response, the instruction-following model can infer the response according to the generated instruction as the max possible. Specifically, we first use the random sampling decoding strategy to generate multiple candidate instructions for one selected response. Then, we concentrate each generated instruction separately with the given response and use the instruction-following model to calculate the perplexity (PPL) of the given response. Finally, we select the generated instruction that causes the lowest PPL score of the given response as the final instruction. Formally, we select the instruction that: $\arg\min_I ppl(O|I)$, where $I$ denotes the candidate instructions and $O$ denotes the selected documents.

## 4.2 EXTRACT-THEN-GENERATE STRATEGY

In the preliminary study, we find that the selected responses have a large effect on the task type of the generated instruction. Therefore, to enrich generated task categories and improve the diversity of the generated instruction data, we propose an extract-then-generate strategy to extract various fragments. Specifically, we explore three strategies to extract fragments: **(1) Keywords**: We use the toolkit Yake [6] (Campos et al., 2018a;b; 2020) to extract the most unique keywords from the original document;**(2) Random Sentence**: We extract one sentence from the original document randomly;**(3) LLM Extraction**: We use the Dolly (Databricks, 2023) to train an extracted model to extract informative and self-contained fragments as the selected response. The template used for the LLM Extraction strategy is:

> ***Below is an instruction that describes a task, paired with an input that provides further context. Write a response that appropriately completes the request.***
>
> ***### Instruction:***
> ***Please extract an informative fragment from the given document.***
>
> ***### Input:***
> ***{Unsupervised Document}***
>
> ***### Response:***

## 5 EXPERIMENT AND ANALYSIS

### 5.1 INSTRUCTION DATASET BASELINES

To conduct a comprehensive comparison, we compare various instruction datasets: **(1) LLaMA-GPT4** (Peng et al., 2023) deftly leverages the formidable power of GPT-4 (OpenAI, 2023) in order to generate responses for the instructions of Alpaca adeptly; **(2) Evol-Instruct** (Xu et al., 2023) put forth some prompt engineering strategies aimed at enhancing the complexity of instructions; **(3) Dromedary** (Sun et al., 2023) propose elaborate prompt engineer strategies to generate instruction data with the use of LLaMa-65B (Touvron et al., 2023); **(4) DYNASAUR** (Yin et al., 2023) leverages the power of LLMs to generate instruction data by harnessing the metadata and data fields of existing NLP datasets; **(5) Dolly** (Databricks, 2023) stands as a remarkable open-source dataset, meticulously composed of human-written instructions that cater to a wide range of general-purpose tasks; **(6) Lima** (Zhou et al., 2023) is a high-quality human-annotated instruction dataset, which is carefully selected manually from large-scale datasets. We also compare with the training split of **(7 ) LongForm** (Köksal et al., 2023)) to compare with in-domain datasets. For LongForm training split, we only use the data generated by the 'text-davinci-003' model from OpenAI API to ensure the fairness of the experiment.

### 5.2 MAIN RESULTS

To compare with various baselines, we report the performance of our framework in Table 1. We can find that our generated instruction data can outperform all the baselines on LongForm. To further confirm the effectiveness of our generated data, we leverage GPT-4 (OpenAI, 2023) to conduct an automatic evaluation on Alpaca Eval (Li et al., 2023; Dubois et al., 2023). The evaluation results are reported in Figure 3, which shows that our generated data can outperform Alpaca slightly. All the evaluation results show the potential of generating high-quality instruction data without using closed-source models.

---

[6]https://github.com/LIAAD/yake

Table 1: The performance of our generated instruction data and all instruction dataset baselines on the LongForm benchmark. ***Bold*** and *Underline* indicate the best and the second best performance respectively. **Potential Risks** denotes whether to use closed-source models to generate instructions, which have potential risks due to the terms of use. Following Köksal et al. (2023), we report the METEOR (Banerjee & Lavie, 2005) score for LongForm. **SE** refers to **StackExchange**; **Wiki** denotes **Wikipedia**; **WH** represents **Wiki-How**; **NI** refers to **Supernaturalinstructions**; **BEA** represents **BEA-GEC**; **BB** denotes **BigBench**.

| Data Source | Potential Risks | All | C4 | SE | Wiki | WH | NI | BEA | Enron | BB |
|---|---|---|---|---|---|---|---|---|---|---|
| LLaMA (Touvron et al., 2023) | ✗ | 18.13 | 10.92 | 18.44 | 19.00 | 10.45 | 17.02 | 61.22 | 8.34 | 17.40 |
| LLaMA-GPT4 (Peng et al., 2023) | ✔ | 19.09 | 14.33 | 16.27 | 24.92 | **17.62** | 18.98 | 43.49 | 9.06 | 10.98 |
| Evol-Instruct (Xu et al., 2023) | ✔ | 20.31 | 13.40 | 16.93 | 25.02 | 17.28 | **19.29** | 61.49 | 9.11 | 12.67 |
| Dolly (Databricks, 2023) | ✗ | 17.92 | 10.24 | 12.65 | 23.76 | 11.71 | 16.29 | 73.35 | 8.41 | 9.45 |
| LongForm (Köksal et al., 2023) | ✗ | 20.27 | **17.34** | 18.22 | **29.50** | 14.60 | 16.84 | 44.62 | **10.29** | 15.78 |
| Lima (Zhou et al., 2023) | ✗ | 21.10 | 13.69 | 19.20 | 22.87 | 15.58 | 18.95 | 71.09 | 8.79 | 17.58 |
| Dromedary (Sun et al., 2023) | ✗ | 19.10 | 10.62 | 13.93 | 29.14 | 10.54 | 16.52 | 74.13 | 8.84 | 16.28 |
| Ours | ✗ | **22.51** | 15.36 | **22.94** | 24.77 | 13.90 | 17.20 | **76.86** | 9.23 | **20.69** |

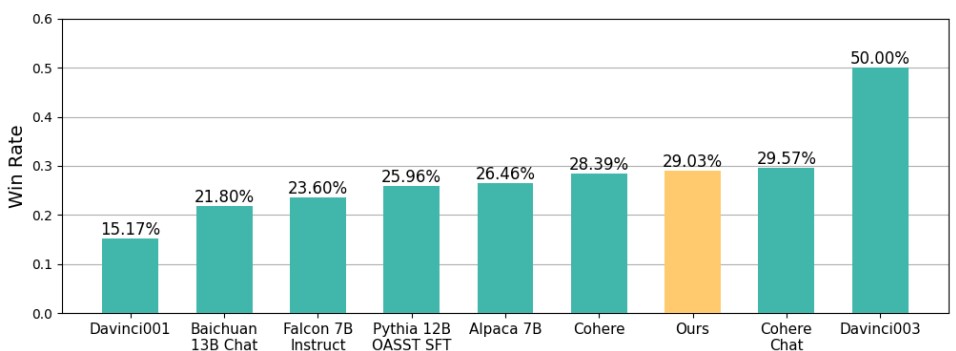

Figure 3: The performance on Alpaca Eval (Li et al., 2023; Dubois et al., 2023). The win rate against Davinci003 is evaluated by GPT-4.

## 5.3 ABLATION STUDY

We also conduct ablation studies to confirm the effectiveness of our proposed strategies, and the results are shown in Figure 4. We can observe that the instruction filtering strategy can bring significant improvements. For the extract-then-generate strategy, we can find that LLM Extraction outperforms Keywords and Random Sentence. Besides, combining origin and LLM Extraction can achieve the best performance, which fully confirms that the extract-then-generate strategy can improve the diversity of instruction data and hence benefit the training of instruction-following models.

## 5.4 INSTRUCTION DIVERSITY ANALYSIS

We also study the diversity of generated instruction data. Following Wang et al. (2022a), we use the Berkeley Neural Parser (Kitaev & Klein, 2018; Kitaev et al., 2019) to extract the top 20 most common root verbs and their top 4 direct noun objects of our generated instruction data, which are shown in Figure 5. From the

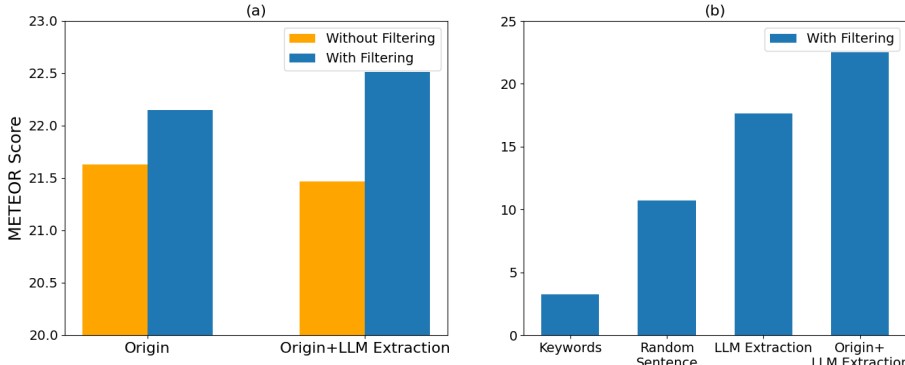

Figure 4: Ablation studies of our proposed strategies. We report the average METEOR score on the Long-Form benchmark. In **(a)**, we study the effect of instruction filtering strategy; In **(b)**, we report the performance of different extract-then-generate strategies. For all instruction generation models, we use Lima as seed instruction data.

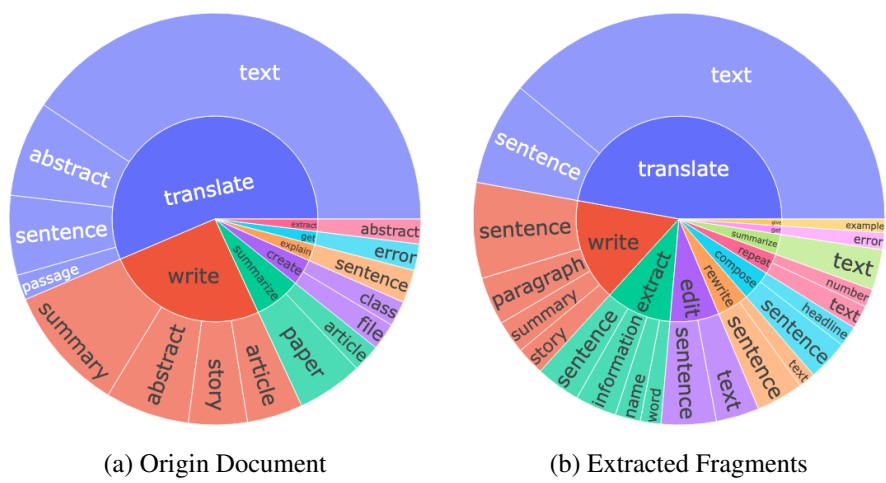

(a) Origin Document         (b) Extracted Fragments

Figure 5: The top 20 most common root verbs (inner circle) and their top 4 direct noun objects (outer circle) in the instruction generated from **(a) Origin Document**; **(b) Extracted Fragments**.

results, we can observe that the extract-then-generate strategy can improve the diversity of the generated instruction data significantly.

## 5.5 OTHER BACKBONE MODELS

To confirm further the effectiveness of our generated data, we also conduct experiments on Baichuan2-7B (Yang et al., 2023a) and BLOOMZ-7B1 (Muennighoff et al., 2022). From Figure 6, we can observe that the instruction-generation strategy can bring significant improvements. Besides, it is surprising that even for the relatively weak models (Baichuan2-7B and BLOOMZ-7B1), the performance of generating

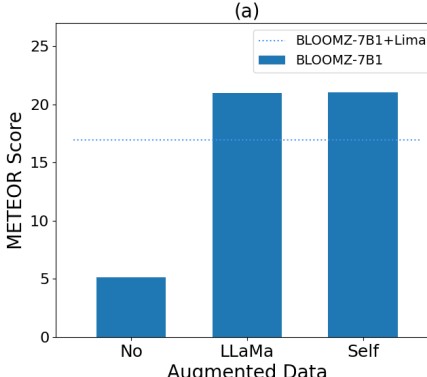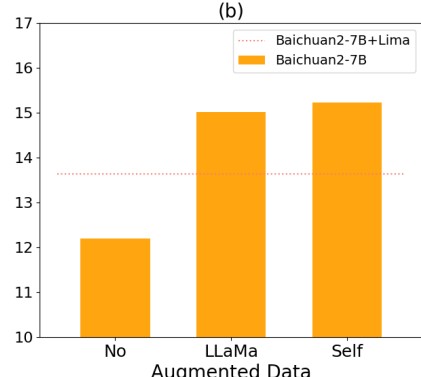

Figure 6: The average performance of other backbone models with different instruction generation strategies on the LongForm benchmark. We compare the performance of not conducting instruction tuning, using seed instruction data for fine-tuning, using self-generated data, and using stronger models for data generation. **(a)** and **(b)** show the training results of BLOOMZ-7B1 and Baichuan2-7B respectively. **'No'** means not conducting instruction tuning, **'LLaMa'** indicates using data generated with LLaMa-7B, and **'Self'** denotes using data generated by the backbone model itself. The dashed line represents the results of the backbone model training with Lima. For instruction generation models, we use Lima as seed instruction data.

data for themselves is on par with distilling from the more powerful model LLaMa-7B, which fully shows the potential of self-alignment. We speculate that these results come from two reasons. First, it is relatively easy to generate instructions for the given response and hence requires lower model capabilities. Second, the instruction-following alignment stage aims to activate the abilities obtained during the pre-training stage, and the model itself knows the best of the knowledge acquired during its pre-training stage. Therefore, the model may generate more appropriate instruction for itself, which can better inspire its own knowledge.

## 6  CONCLUSION

Instruction tuning has gained more and more attention in recent years, which uses high-quality instruction data to enable LLMs to generalize to various tasks. In this work, to generate high-quality instruction data automatically and avoid the potential risks of using closed-source models to generate data, we systematically explore using open-source models to generate high-quality instruction datasets for themselves in a semi-supervised method. We find that the instruction-generation strategy can outperform the self-training strategy significantly. We speculate that it is because that the difficulty of generating instruction is lower and the instruction-generation strategy can ensure the validity of responses. Therefore, to further improve the diversity of instructions and alignment between instructions and responses generated by the instruction-generation strategy, we propose the instruction filtering strategy and extract-then-generate strategy. Both LongForm benchmark and GPT-4 evaluation confirm the effectiveness of our proposed strategies. Furthermore, we find that with the help of our proposed strategies, the relatively weak models, the performance of generating data for themselves is on par with distilling from the more powerful model. Despite the promising results achieved in this work, this work is a preliminary study to explore the potential of using open-source models to generate high-quality instruction datasets for themselves. We hope more progress can be achieved in the self-alignment and this work can prompt the development of open-source instruction tuning models to continuously narrow the gap with closed-source models.

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
