# OpenReview forum: "SASS: Self-Alignment with Semi-Supervised Instruction Data Generation"
_ICLR.cc/2024/Conference — Submitted to ICLR 2024_

### Official Review · Reviewer_ZkP4 · 2023-10-28

**Soundness:** 2 fair
**Presentation:** 2 fair
**Contribution:** 2 fair
**Rating:** 3
**Confidence:** 3

**Summary:**

This paper proposes *Sass*, a framework for instruction data generation using open-source models. The paper starts by exploring two main semi-supervised generation settings: generating responses for existing instructions (self-training) and generating instructions for responses (instruction generation). The authors found that the instruction-generation strategy is better. Based on this observation, this paper further proposes an instruction-filtering strategy and extract-then-generate strategy to ensure the alignment and diversity of instructions. With the constructed instruction-tuning data, experiments are conducted on LongForm and Alpaca Eval, both indicating better performance.

**Strengths:**

- This paper focuses on the curation of the instruction-tuning dataset, which is an interesting topic for the community.
- The general writing starts from a preliminary investigation, which helps to understand the motivation and provides some insights.

**Weaknesses:**

- It's hard to position the paper in the research line of instruction-tuning data curation. There are many existing works focusing on either generating instructions given responses or generating responses given instructions, as introduced in the related work section. However, the authors did not discuss the relatedness or differences compared with previous works, which makes it even harder to understand the contribution of this work. Also, this paper forgets an important baseline [1].
- Although I like the preliminary study, the analysis in Section 3 is not sound and detailed enough. For example, why use more Dolly data leads to worse performance? Why Lima + Dolly together is even worse? How do you arrive at the conclusion that "this phenomenon further confirms our hypotheses that it is easier to generate appropriate instructions for the given response."?
- The experiments and evaluations are not so convincing. The proposed method *Sass* is just evaluated on two automatic evaluation benchmarks. It is necessary to have human evaluation and examine the quality of the generated instructions of multiple dimensions. Also, the performance is not that significant compared with previous baselines.
-  The experiment details for results in Section 5 are missing.

[1] Li, Xian, et al. "Self-alignment with instruction backtranslation." arXiv preprint arXiv:2308.06259 (2023).

**Questions:**

- I don't quite understand the "extract-and-then-generate" method. What does "keywords" and "random sentence" mean? Is it only using keywords or randomly extracting one sentence as the response?
- How Dolly is used for training LLM Extraction? What is the input and the response?

---

### Official Review · Reviewer_d3RB · 2023-10-31

**Soundness:** 2 fair
**Presentation:** 2 fair
**Contribution:** 2 fair
**Rating:** 3
**Confidence:** 5

**Summary:**

The paper discusses the significance of instruction tuning for Large Language Models (LLMs) to meet user expectations in diverse open-domain tasks. It emphasizes the necessity of high-quality instruction data, which is challenging to obtain due to the high costs of human annotation. The paper explores utilizing open-source models to autonomously generate instruction data, as opposed to leveraging powerful closed-source models. It introduces semi-supervised instruction data generation strategies to enhance the performance of open-source models. The authors propose two techniques, namely instruction filtering and an extract-then-generate strategy, to augment the diversity and alignment of generated instructions. Through experiments on the LongForm benchmark and GPT-4 evaluation, the strategies demonstrate effectiveness. Interestingly, even weaker models self-generating data exhibit performance comparable to distillation from stronger models, showcasing potential for progress in generating quality instruction data without relying on closed-source models.

**Strengths:**

1. The paper proposes a self-alignment with semi-supervised instruction data generation method to improve the instruction tuning.
2.  The method performs well on LongForm benchmark.

**Weaknesses:**

1. Lack of novelty: the self-alignment is not a new idea that has already been proved its effectiveness on RLAIF[1] and Constitutional AI[2]. This paper only proposes a prompt to do the self-alignment which lacks novelty.

2. Lack of experiment on larger models: only running the experiment on LLaMA-7B models is very weak and it cannot prove that the proposed methods can scale up to larger scale models. I suggest the authors run experiments on 13B and 33B models as well. Moreover, the authors should also try LLaMA-2-7B as their base model to fine-tune.

3. Lacks of discussion of many relevant works on data filtering:
a. Instruction mining: High-quality instruction data selection for large language models. Cao et al., 2023
b. InstructionGPT-4: A 200-Instruction Paradigm for Fine-Tuning MiniGPT-4. Wei et al., 2023
d. AlpaGasus: Training A Better Alpaca with Fewer Data. Chen et al., 2023
Missing discussion with the self-alignment papers:
a. Self-Alignment with Instruction Backtranslation. Li et al., 2023

Before claiming any novelties in filtering, please discuss the difference between your methods with these papers.
4. Missing experiments: Alpaca-eval leaderboard.

[1] RLAIF: Scaling Reinforcement Learning from Human Feedback with AI Feedback. Google
[2] Constitutional AI: Harmlessness from AI Feedback. Anthropic.

**Questions:**

1. Could you please show some results on the alpaca-eval leaderboard? (very important and widely used leaderboard  to measure the instruction-following abilities of the models)
2. Could you please show the results on OpenLLM leaderboard? ( it is a leaderboard to measure the benchmark performance)

---

### Official Review · Reviewer_jPLq · 2023-11-01

**Soundness:** 3 good
**Presentation:** 2 fair
**Contribution:** 2 fair
**Rating:** 5
**Confidence:** 3

**Summary:**

The paper explores alternative methods to create high-quality instruction data without depending on closed-source models, presenting strategies to improve data quality. The authors investigate different established methods for generating instructions, incorporating the most effective version along with two innovative approaches to further improve quality. Evaluation results demonstrate the effectiveness of this approach when compared to Alpaca and GPT-4.

**Strengths:**

This paper provides a simple method to use open-source LLMs to generate instruction-tuning data. It is well motivated by the IP issues that exist in deploying company's own instruction-tuned models. Experimental results show that SASS can harvest better performance than many other instruction-tuning data including GPT-generated and human-curated ones. The ablation studies help us to better understand the effect of filtering not suitable instructions.

**Weaknesses:**

Despite the good performance, there still leave several weaknesses:

**Lack of comparison with other possible methods based on Llama-2**: For instance, if one has access to Llama-2-70B-chat or even smaller versions like LLAMA-2-13B-chat, it raises the question of whether it's feasible to directly employ the Self-Instruct method to instruct the model to generate new instructions. This alternative could obviate the need for extensive training of open models to perform tasks involving instruction generation or output generation. There's a need for comprehensive exploration of other potential methods for generating instruction-tuning data based on open models.

**Novelty limitations**: It leaves me an impression that the paper are actually doing fundamentally very similar works such as instruction -> output (Instruction GPT-4) and output -> instruction (Dynosaur and Instruction Induction (Honovich et al., 2022)). But the models are simply changed from GPT-series models to open-source LLMs. It might be better to see if it is possible to invent a brand new paradigm for generating the instruction-tuning data instead of mainly adopting the previous formulations.

**Questions:**

1. What is the Dynosaur performance in Table 1?

---

### Official Review · Reviewer_oFCh · 2023-11-01

**Soundness:** 2 fair
**Presentation:** 2 fair
**Contribution:** 2 fair
**Rating:** 3
**Confidence:** 4

**Summary:**

This work introduces SASS, a semi-supervised instruction generation method with open-source models. To create the training dataset for instruction generation, they propose to flip the input-output pairs from existing instruction tuning datasets. Then, at test time, they generate instructions from unlabeled or unsupervised corpus to produce instructions to augment the existing training dataset and train another model. The results on the LongForm benchmark show that the proposed method, SASS, outperforms existing methods on average. Finally, the ablations show that the same method can be extended to other models such as BLOOMZ and Baichuan-7B.

**Strengths:**

**Good problem and motivation.**
The paper is well-motivated and timely. Existing work mostly focuses on instruction generation with closed-source API-based models which cannot be used for further training models. This work shows that openly available models can be used to generate instructions.

**Interesting method.**
The proposed method of flipping the input-output pairs is interesting and shows positive results on the downstream benchmark tasks. Further, the authors introduce the extract-then-generate strategy that categorizes the generations at three levels. I found this part of the paper to be very organized.

**Perplexity as filtering.**
While existing work has used GPT-based models to clean the generated instructions, the work shows that perplexity is a viable alternative to filtering datasets.

**Weaknesses:**

**Need more evaluation.**
- While the proposed work improves over the LongForm benchmark, more evaluation is needed. What are the reasons for the improvement in performance compared to the baselines? For example, Dromedary underperforms SASS on the LongForm benchmark despite being trained on more instructions and a larger model (Dromedary 65B and SASS 7B). Is there a difference in the training dataset that causes this discrepancy? If so, would it be possible to train Dromedary 7B on your corpus (minipile)?

- The work also misses out on providing qualitative results, i.e., the generations from the model. How do the generations from the model compare to GPT-4 like models? How do they differ? Are there types of tasks that the quality of the generations is better?

**Discrepancies in the numbers reported from baseline papers.**
The average METEOR score reported by Koksal et. al. [a], the main baseline in the paper, is 24.8 (see Table 2 in [a]). But, in Table 1, this work reports 20.27 which is significantly lower than the baseline. Since the proposed work reports a METEOR score of 22.51, I’d be curious to know the reason for the discrepancy.

**Missing key implementation details.**
Some of the details regarding the baselines are missing. LIMA [b] does not release their dataset to the public instead they mention the sources of the dataset. It would be great if the authors could provide more details regarding the approximations they made when creating the LIMA training dataset. Furthermore, [b] also trains their model with a different learning rate and uses linear decay. It isn’t clear from the paper if the work has changed any of these hyperparameters.

**Paper organization and writing is not clear.**
- The preliminary study could be moved after the method section because several details from the method section are required to understand the preliminary study. Furthermore, Figure 1 could be improved by monotonically increasing the amount of augmented data. At the moment, the x-axis reads as follows Dolly 1000 -> Dolly 5000 -> Dolly 14K -> LIMA 1000 -> Dolly 14K + LIMA 1000). Finally, it is unclear why the work omits results on self-training with Dolly 1000 and Dolly 5000.

- The paper introduces the term “origin document” when analyzing the extracted fragments without sufficient content/background. It would be helpful for the readers to standardize the terms throughout the paper.

**Related work.**
I’d like to point out a few related and concurrent works that have not been included in the related work section:
- Self-Alignment with Instruction Backtranslation. ArXiV 23.
- Learning Instructions with Unlabeled Data for Zero-Shot Cross-Task Generalization. EMNLP 22.

**Nit:**

Page 2, paragraph 1: tight alignment -> alignment (because I’m not sure what tight alignment means as opposed to loose alignment)

DYNASAUR missing from Table 1.

**References**

[a] LongForm: Optimizing Instruction Tuning for Long Text Generation with Corpus Extraction. ArXiV 23.

[b] LIMA: Less Is More for Alignment. ArXiV 23.

**Questions:**

Please find the questions in the weaknesses.

---

### Meta-Review · Area_Chair_onPE · 2023-12-05

**Metareview:**

This paper studies self-alignment of LLMs with semi-supervised instruction data generation. All reviewers consistently have significant concerns on the novelty, writing, experiments of the paper, and the authors did not respond to these concerns in the rebuttal phase. AC hopes the reviews are helpful for the authors to prepare for a next version of the paper.

**Justification For Why Not Higher Score:**

The authors did not provide any rebuttal in response to the reviewers' concerns. The paper is clearly below the bar according to the reviewers' comments and scores.

**Justification For Why Not Lower Score:**

N/A

---

### Decision · Program_Chairs · 2024-01-16

Reject